# Expanded transcriptomic view of strawberry fruit ripening through meta-analysis

**Gibum Yi** [1]ᵒ*, **Hosub Shin**[2]ᵒ, **Kyeonglim Min**[2], **Eun Jin Lee**[2,3]*

**1** Department of Bio-Environmental Chemistry, College of Agriculture and Life Sciences, Chungnam National University, Daejoen, Korea, **2** Department of Agriculture, Forestry and Bioresources, College of Agriculture and Life Sciences, Seoul National University, Seoul, Korea, **3** Research Institute of Agriculture and Life Sciences, Seoul National University, Seoul, Korea

ᵒ These authors contributed equally to this work.
* gibumyi@cnu.ac.kr (GY); ejinlee3@snu.ac.kr (EJL)

**Data Availability Statement:** All material files are available from the https://www.ncbi.nlm.nih.gov/sra database (PRJNA394190, RJNA552213, PRJNA564159).

## Abstract

Strawberry is an important fruit crop and a model for studying non-climacteric fruit ripening. Fruit ripening and senescence influence strawberry fruit quality and postharvest storability, and have been intensively studied. However, genetic and physiological differences among cultivars preclude consensus understanding of these processes. We therefore performed a meta-analysis by mapping existing transcriptome data to the newly published and improved strawberry reference genome and extracted meta-differentially expressed genes (meta-DEGs) from six cultivars to provide an expanded transcriptomic view of strawberry ripening. We identified cultivar-specific transcriptome changes in anthocyanin biosynthesis-related genes and common changes in cell wall degradation, chlorophyll degradation, and starch metabolism-related genes during ripening. We also identified 483 meta-DEGs enriched in gene ontology categories related to photosynthesis and amino acid and fatty acid biosynthesis that had not been revealed in previous studies. We conclude that meta-analysis of existing transcriptome studies can effectively address fundamental questions in plant sciences.

## Introduction

Strawberry (*Fragaria × ananassa* Duch.) is an important and nutritious fresh fruit crop for human consumption. As its popularity has increased, so have research and breeding efforts. Strawberry has an octoploid origin resulting from the combination of four diploid species, *F. iinumae*, *F. nipponica*, *F. viridis*, and *F. vesca*, and this genomic complexity has made genetic and genomic studies inefficient until recently [1].

Ripening is a complex process integrating development and senescence. Fruits can be divided into two types, climacteric and non-climacteric, based on their respiration and ethylene fluctuation during ripening. Climacteric fruit such as tomato have a respiratory burst and ethylene peak at the onset of ripening and have been studied intensively. By contrast, non-climacteric fruit produce a very small amount of ethylene with no increased rate of respiration. Strawberry is used as a model for non-climacteric fruit ripening because of its commercial importance and experimental advantages [2]. Expressed sequence tag (EST)-based transcript

**Funding:** This work was supported by the Basic Science Research Program through the National Research Foundation (NRF, 2016R1A1A1A05919210) of Korea funded by the Ministry of Education, Science, and Technology, the Rural Development Administration (RDA, PJ01364804), and the Korea Institute of Planning and Evaluation for Technology in Food, Agriculture, and Forestry (IPET, 617068-05-1-WT111), Republic of Korea. And the recipient of these three funds is Dr. Eun Jin Lee. The funders had no role in study design, data collection and analysis, decision to publish, or preparation of the manuscript.

**Competing interests:** The authors have declared that no competing interests exist.

analysis, microarray analysis, and recent next-generation sequencing (NGS)-based transcriptome analysis have been performed to understand the molecular mechanisms of strawberry ripening [3–5], with metabolic and proteomic analysis supporting this effort [6–8]. Physiological changes during ripening affect texture, acidity, color, flavor, and aroma, occurring alongside molecular changes in plant hormone signaling, cell wall loosening, sugar transport, and anthocyanin biosynthesis [6–9].

Strawberry is asexually propagated by runners, and breeding efforts focus on selecting superior plants rather than establishing inbred lines. Intensive breeding programs have been implemented since the 1990s in many countries; dominant cultivars differ by region and are genetically diverse, with complex relationships between cultivars [10]. Research in strawberry has been performed using a wide range of cultivars. Understanding the genetic relationships among current cultivars is therefore necessary for combined analysis. The recently published, chromosome-based genome of cultivar 'Camarosa' enables comparison of previously published transcriptome data to a common reference genome [1].

Despite accumulating data showing transcriptome changes during strawberry development, the effects of the genetic backgrounds in different cultivars has not been investigated; thus, it is difficult to conclude whether changes are cultivar-specific or not. Furthermore, use of different reference genomes or *de novo* assembled transcriptomes along with different gene names means that data from the same developmental stage or obtained under similar conditions are not easily comparable.

Meta-analysis is being applied in the plant science field to provide an expanded view of specific biological questions that cannot be answered in a single experiment [11, 12]. However, application of meta-analysis is relatively limited in plant biology compared to that in medical science or environmental sciences. There are likely two main reasons for this. First, it is easier to produce a randomized sample design and high enough number of biological replicates using plant samples compared to animal or human samples. Second, identical treatments are imposed much less frequently in plant science studies than in medical science. However, even in plant science research, transcriptome data is produced from a relatively small number of biological replicates, generally two to three, owing to the cost per sample. Meta-analyses have the potential to increase the usefulness of studies involving a limited number of samples by compiling multiple studies, as in the application of multiple transcriptome data sets to elucidate concordant changes by identifying meta-DEGs (differentially expressed genes) [11, 12]. Furthermore, meta-analysis can be applied to RNA sequencing (RNA-Seq) data from different species, or under different biotic and abiotic stresses, to address a broad range of questions [13, 14].

The recently published, high-quality genome of strawberry, which has 805 Mb of sequence, covering the 28 expected chromosome-level pseudomolecules [1], allows meta-analysis to be used for systematic comparison of different studies. Here, we compiled previous transcriptome data from various cultivars grown under different conditions and present an expanded transcriptomic view of ripening in strawberry. We used meta-analysis of transcriptome data for six cultivars from three independent studies to deliver an in-depth view of the results, which provide further information on strawberry ripening to the research community.

## Methods

### Transcriptome data analysis and meta-analysis

Data from three publicly available transcriptome studies were used for meta-analysis (**Table 1**). The transcriptome data include the whole-fruit transcriptomes of six different cultivars at two coinciding developmental stages, Big Green (BG) and Fully Red (FR). RNA-Seq raw data were downloaded from the NCBI SRA database and low-quality reads (Q < 20) were

**Table 1. Public transcriptome data used in this study.**

| Project number* | Cultivar | Developmental stages | Reps | Sequencing platform | Raw reads (×1000) | Reference |
|---|---|---|---|---|---|---|
| PRJNA 394190 | Toyonoka | Large green, Red | 2 | HiSeq × Ten | 240,550 | Hu *et al.* 2018 |
| PRJNA 552213 | Benihoppe | Green, Full red | 2 | HiSeq 4000 | 241,181 | NA |
| | Xiaobai | Green, Full red | 2 | HiSeq 4000 | 219,467 | NA |
| | Snow princess | Green, Full red | 2 | HiSeq 4000 | 220,347 | NA |
| PRJNA 564159 | Sunnyberry | Big green, Fully red | 3 | HiSeq 4000 | 310,134 | Min *et al.* 2020 |
| | Kingsberry | Big green, Fully red | 3 | HiSeq 4000 | 331,885 | |

*NCBI BioProject accession.

Reps, Number of biological replications.

NA, not available.

filtered out using FASTX-Toolkit. Filtered reads were mapped to the *Fragaria × ananassa* 'Camarosa' Genome Assembly v1.0 (https://www.rosaceae.org) using Tophat v2.1.1 [15] with default parameters, and the number of mapped reads was counted using ht-seq-count from HTSeq [16]. For single analyses, differentially expressed genes (DEGs) were identified using the Bioconductor package edgeR 3.30.3 [17] with minimum FPKM (Fragments Per Kilobase of transcript per Million fragments mapped) $> 0.3$, false discovery rate (FDR) $< 0.05$, and log2 fold change $> 1$. Meta-analysis was performed following the method of Cohen *et al.* [18] with minor modification. Briefly, p-values from single analyses were combined using the Fisher's sum of logs method using the R package metap v1.1 [19] and multiple tests were performed using the p.adjust function in R with the 'fdr' method. Meta-DEGs were identified with median FPKM $> 0.3$ and absolute value of median log2 fold changes $> 1$ for all cultivars within the analysis, and an adjusted p-value $< 0.01$.

Principal component analysis (PCA) was performed using the R package FactoMineR 1.32 [20] with the whole-transcriptome FPKM values of all samples from six cultivars.

## Gene ontology (GO) term and KEGG pathway enrichment analysis

Fisher's exact test was performed using TopGo 2.18 [21] in the R package for GO term enrichment tests. Adjusted p-values were calculated using the p.adjust function in R with the 'fdr' method, and significantly enriched GO terms were identified with FDR $< 0.05$. Enrichment test for KEGG (Kyoto Encyclopedia of Genes and Genomes) [22] pathways was performed using Fisher's exact test, and p-values were adjusted using the FDR method. Significantly enriched pathways were determined with FDR $< 0.05$ and odds ratio $> 1$.

## Single-nucleotide polymorphism (SNP) detection and genetic relationship analysis

Filtered reads were mapped to the *Fragaria × ananassa* 'Camarosa' Genome Assembly v1.0 (https://www.rosaceae.org) using Burrows-Wheeler aligner v0.7.17-r1188 [23] with the 'mem' algorithm. SAMtools v0.1.19 [24] was used for calling variant and homozygous SNPs for all six cultivars covered by $\geq 3$ reads per sample. A total of 7,002 SNPs were identified, and a neighbor-joining tree was reconstructed using MEGA X [25].

## Motif search for the meta-DEGs

*De novo* motif analysis was performed with 1 kb upstream sequence of the transcription start site of META DEGs using MEME software with "-revcomp -mod zoops -objfun de"

parameters [26]. The second-order Markov background model was constructed based on upstream sequences of all genes and the model was used as control. Enriched motifs were filtered with E-value threshold of 0.05. The possible binding site of transcription factor was searched using TOMTOM [27] with JASPAR CORE (plant) database 2018 [28].

# Results

## The six cultivars have distinct characteristics

During the BG to FR stages, strawberry fruit undergoes de-greening and red coloration among other physiological changes [5]. The two stages are visually separated from the other developmental stages and have been selected in many previous studies investigating the ripening process [29, 30]. More DEGs are generated from BG vs. FR than from sub-stages between BG and FR [29].

For meta-analysis of strawberry fruit ripening, we collected publicly available RNA-Seq data for BG and FR stages, for which the largest dataset is available (Table 1). We characterized the six cultivars based on their RNA-Seq results since the meta-analysis assumed that the six cultivars are distinct from each other and have the same weight for analysis. PCA based on their fruit transcriptome profiles at the two developmental stages, BG and FR, showed clustering of samples where PC1 and PC2 explained 36% of the variation. The two different developmental stages were relatively well separated by PC1 and PC2. Furthermore, there seemed to be two clusters of cultivars at both developmental stages. 'Kingsberry' and 'Sunnyberry' were clustered together and the other cultivars were clustered with each other. Most of the replicates were closely located, with those of 'Toyonoka' and 'Sunnyberry' at the BG stage showing a greater distance (Fig 1A). Transcriptome profiles tended to show greater variation at the BG

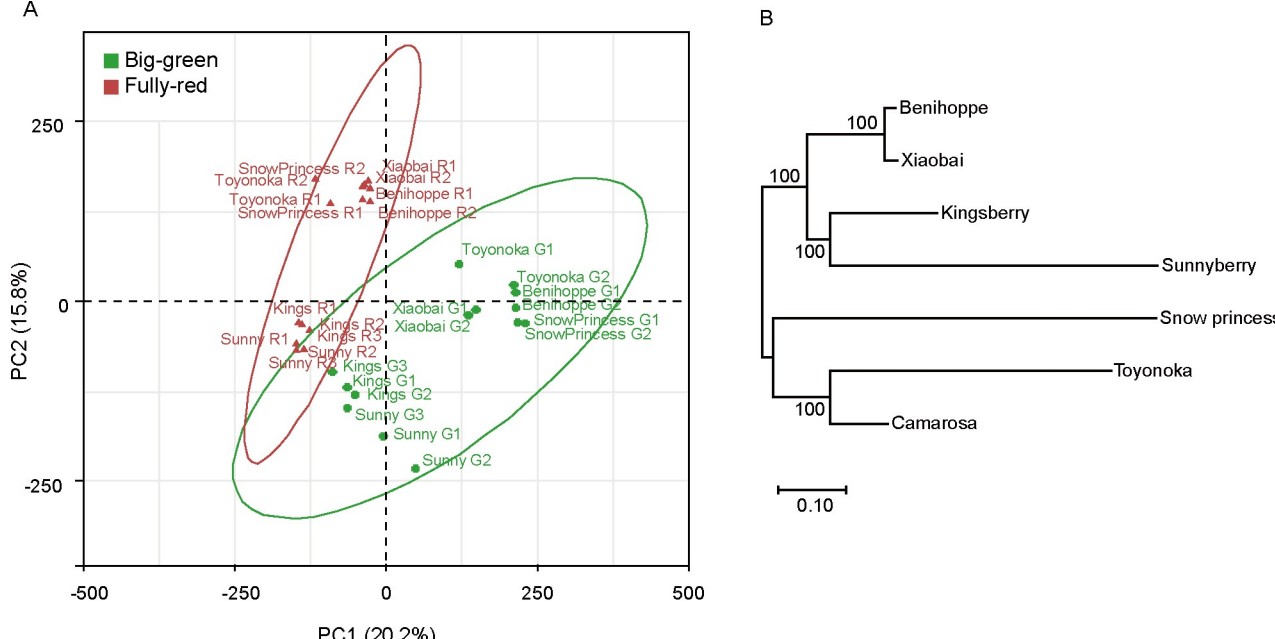

**Fig 1. Transcriptomic characteristics of the six strawberry cultivars.** (A) PCA based on transcriptome profiles. FPKM values for all genes were scaled by unit variance with the R package FactoMineR. Samples from the two developmental stages, BG and FR, are colored green and red, respectively. Areas bounded by green and red lines indicate 95% confidence area for BG and FR, respectively. (B) Genetic distance among the six cultivars and the reference cultivar 'Camarosa' based on 7,002 SNPs. Numbers on branches indicate percentage bootstrap support from 1,000 replications. Bar, nucleotide substitution rate for the SNP loci.

stage compared to the FR stage, as determined from the distribution of samples on the PCA score plot (**Fig 1A**), and as observed in a previous study using 'Kingsberry' and 'Sunnyberry' [30].

We further compared cultivars based on SNPs in genic sequences using the transcriptome data. 'Xiaobai' was first introduced from a somatic variant of 'Benihoppe' [4], and as expected the genetic distance of these two cultivars was small (**Fig 1B**). However, their transcript profiles at the BG stage were distinct enough to consider them different cultivars. 'Kingsberry' and 'Sunnyberry' showed a relatively close genetic relationship from the SNP genotypes (**Fig 1B**); these cultivars are from the same breeding institute and share partial ancestry [30]. 'Kingsberry' also showed a relationship to 'Benihoppe' and 'Xiaobai' (**Fig 1B**), which share the maternal parent 'Akihime'. 'Toyonoka' showed similarity to the reference cultivar 'Camarosa'. These cultivars were genetically diverse to cover certain amount of genetic diversities of strawberry [4, 30]. From these analyses, we concluded that there was sufficient genetic variation among these six samples to consider them separate cultivars and perform further analyses.

## Reanalysis of publicly available strawberry transcriptome data

The mapping rate for 'Kingsberry' and 'Sunnyberry' was slightly improved to 66.5%, compared to 63.6% using the old reference genome [30, 31], showing the improvement in the reference genome (**S1 Table**) [1]. Furthermore, there was no significant difference in mapping rate among cultivars analyzed in this study, indicating similar quality of RNA-Seq data (**S1 Table**). The reference genome contains 108,087 protein-coding genes [1], among which 44,061 (40.8%) genes were not transcribed (FPKM < 0.3) in any of the samples of the six cultivars and 11,968 (11.1%) genes were expressed only at a basal level (FPKM < 1) in only one of the samples.

We analyzed correlation coefficients ($R^2$) among biological replicates and among cultivars (**S2 Table**). The average $R^2$ among biological replicates was 0.939, ranging from 0.868 to 0.998, showing high concordance among replicates. The highest correlations between cultivars were observed for 'Xiaobai' FR and 'Benihoppe' FR samples. Notably, the 'Sunnyberry' BG sample showed higher correlation to the FR samples of 'Benihoppe', 'Xiaobai', and 'Snow princess' than to the BG samples.

We identified more DEGs between FR and BG samples than previous studies; for example, 'Sunnyberry' had 4,656 DEGs between these two stages in a previous study [30] but we detected 10,033. There were 1.5–2.4 times more DEGs down-regulated in FR compared to BG than up-regulated in the six cultivars. Different reference genomes and criteria can easily change the number of DEGs detected; however, the ratio between up- and down-regulated DEGs was consistent across data sets (**Fig 2A**). Numbers of up- and down-regulated DEGs in FR compared to BG in the six cultivars are shown as Venn diagrams (**Fig 2B and 2C**). 'Toyonoka' and 'Snow princess' had the largest number of cultivar-specific down- and up-regulated DEGs, respectively. 'Xiaobai' and 'Benihoppe' had the largest number of two-cultivar-specific up-regulated DEGs, whereas 'Toyonoka' and 'Snow princess' shared the largest number of two-cultivar-specific down-regulated DEGs (**Fig 2B and 2C**). Over half (51.5%) of the up-regulated DEGs were single-cultivar specific whereas only 35.2% of the down-regulated DEGs were single-cultivar specific (**Fig 2B and 2C**). By contrast, 45.3% of the down-regulated DEGs were commonly detected in three or more cultivars, whereas only 27.8% of the up-regulated DEGs were commonly detected in three or more cultivars (**Fig 2B and 2C**). Down-regulated DEGs tended to be common to several cultivars whereas up-regulated DEGs tended to be cultivar specific.

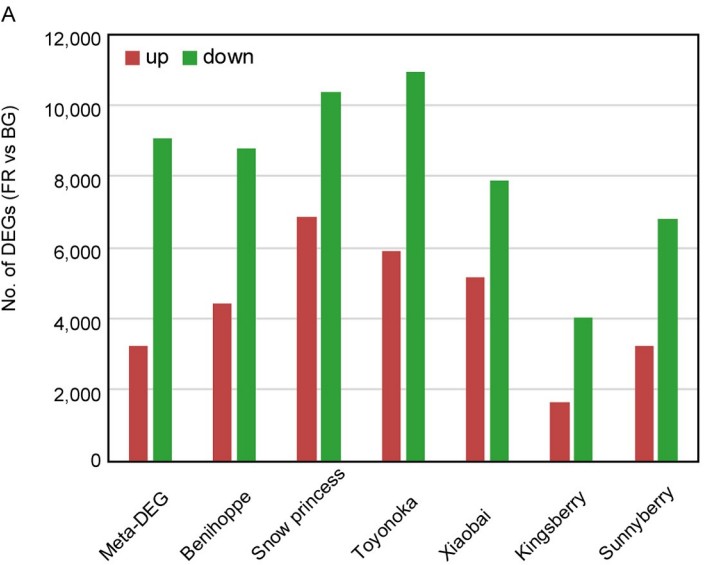

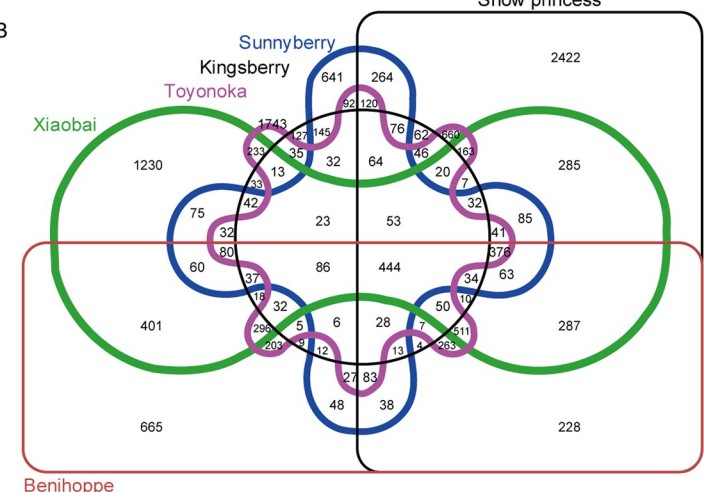

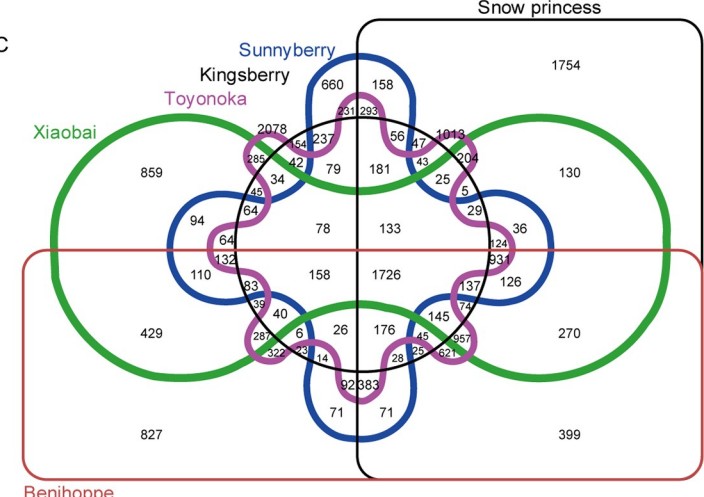

**Fig 2. Meta-DEGs and DEGs from the six strawberry cultivars.** (A) Numbers of up- and down-regulated DEGs between big green (BG) and fully red (FR) stages are shown as red and green bars, respectively. (B) Venn diagram of the up-regulated DEGs (FR vs. BG) in the six cultivars. Numbers in each section represent the specific or common DEGs among six cultivars. (C) Venn diagram of the down-regulated DEGs (FR vs. BG) in the six cultivars.

We applied GO enrichment analysis to the DEGs of the six cultivars to identify functions important during ripening (**Tables 2 and 3**). Down-regulated DEGs in the six cultivars shared common GO terms whereas up-regulated DEGs had few common terms. 'photosynthesis' (GO:0015979) and 'cell wall biogenesis' (GO:0042546) were the two most highly enriched GO terms among down-regulated DEGs for all six cultivars. Other terms related to cell wall loosening and sugar metabolism were commonly associated with DEGs from all six cultivars (**Tables 2 and 3**).

Cultivar specificity was more apparent in GO analysis of DEGs up-regulated at the FR compared to BG stage. 'Benihoppe', 'Xiaobai', and 'Snow princess' showed similar GO term enrichment whereas 'Sunnyberry' and 'Kingsberry' shared a distinct set of enriched GO terms. We also observed these differences among cultivars in the PCA clustering (**Fig 1**). Fewer up-regulated DEGs than down-regulated DEGs and less-conserved GO term enrichment among the up-regulated DEGs indicate that strawberry ripening from the BG to FR stages is active having a large number of genes with decreased expression. Furthermore, these ripening processes are quite variable among different cultivars.

## Meta-analysis reveals expanded transcriptome changes during ripening

We identified 12,339 meta-DEGs by meta-analysis of the six cultivars consisting of 3,248 up- and 9,091 down-regulated DEGs in FR compared with BG samples (**Fig 2A**). As expected from individual analysis of the six cultivars, there were 2.8 times more down-regulated meta-DEGs than up-regulated meta-DEGs. Between 54.5 and 79.0% of DEGs in the six cultivars were retained as meta-DEGs (**S3 Table**).

The highly enriched GO terms associated with meta-DEGs were 'organic acid metabolic process' (GO:0006082) and 'fatty acid metabolic process' (GO:0006631). However, these terms were enriched only in two or three of the cultivars, respectively. For DEGs down-regulated at FR compared to BG stage, 'photosynthesis' (GO:0015979) and 'cell wall biogenesis' (GO:0042546) were the two most highly enriched GO terms among meta-DEGs, as for the six cultivars (**Table 3**). Other terms related to cell wall loosening such as 'cell wall polysaccharide metabolic process' (GO:0010383) and 'cellular carbohydrate metabolic process' (GO:0044262), which were associated with DEGs from all six cultivars, were also enriched among the meta-DEGs. KEGG analysis revealed similar findings to GO analysis, with 'photosynthesis,' 'glyoxylate and dicarboxylate metabolism,' 'cutin, suberine, and wax biosynthesis,' and 'starch and sucrose metabolism' being enriched pathways. We identified clear, common changes in starch metabolism, chlorophyll degradation, and cell wall degradation during the ripening process in all six cultivars.

Of the meta-DEGs, 2,150 were detected in all six cultivars, 695 genes were differentially expressed in only one of the six cultivars, whereas 483 were not detected as DEGs in any of the six individual cultivars (**S4 Table**). These 483 meta-DEGs were assumed to be novel DEGs. When GO enrichment analysis was performed on these novel DEGs, 'photosynthesis, light reaction' (GO:0019684), 'cellular modified amino acid biosynthetic process' (GO:0042398), and 'fatty acid biosynthetic process' (GO:0006633) terms were associated with DEGs down-regulated in FR compared with BG samples, and 'response to auxin' (GO:0009733) was associated with up-regulated DEGs. These newly detected meta-DEGs provide an extended view of

**Table 2. Enriched GO terms for up-regulated meta-DEGs and FDR values for up-regulated DEGs (FR vs. BG) in the six cultivars.**

| GO id | Term | GO level | Annotated | Assigned | Expected | FDR | | | | | | |
|---|---|---|---|---|---|---|---|---|---|---|---|---|
| | | | | | | Meta | Toyo | Beni | Xiao | Snow | Sunny | Kings |
| 0006082 | Organic acid metabolic process | 4 | 649 | 52 | 24.23 | 4.5E-05 | 0.1510 | 2.9E-09 | 3.6E-20 | 3.0E-05 | 0.0388 | 1 |
| 0043436 | Oxoacid metabolic process | 5 | 649 | 52 | 24.23 | 4.5E-05 | 0.1510 | 2.9E-09 | 3.6E-20 | 3.0E-05 | 0.0388 | 1 |
| 0006631 | Fatty acid metabolic process | 5 | 59 | 13 | 2.2 | 6.3E-05 | 0.0060 | 1.4E-05 | 1.1E-06 | 0.0238 | 0.0411 | 0.0691 |
| 0044255 | Cellular lipid metabolic process | 4 | 200 | 24 | 7.47 | 0.0001 | 0.0001 | 4.6E-05 | 7.2E-08 | 0.0130 | 0.0551 | 0.0552 |
| 0030329 | Prenylcysteine metabolic process | 5 | 4 | 4 | 0.15 | 0.0002 | 0.0017 | 0.0002 | 0.0003 | 0.0035 | 1 | 1 |
| 0042138 | Meiotic DNA double-strand break formation | 5 | 4 | 4 | 0.15 | 0.0002 | 0.0017 | 0.0002 | 0.0003 | 0.0035 | 0.0004 | 0.0003 |
| 0044273 | Sulfur compound catabolic process | 5 | 4 | 4 | 0.15 | 0.0002 | 0.0017 | 0.0002 | 0.0003 | 0.0035 | 1 | 1 |
| 0005975 | Carbohydrate metabolic process | 4 | 391 | 33 | 14.6 | 0.0009 | 0.0613 | 0.0022 | 3.3E-05 | 0.5840 | 0.0024 | 0.0691 |
| 0001678 | Cellular glucose homeostasis | 5 | 16 | 6 | 0.6 | 0.0012 | 0.0160 | 0.0022 | 0.0051 | 0.2046 | 0.0021 | 1 |
| 0017144 | Drug metabolic process | 4 | 279 | 26 | 10.41 | 0.0012 | 0.4706 | 7.1E-06 | 4.3E-18 | 7.5E-05 | 0.0142 | 1 |
| 0006629 | Lipid metabolic process | 4 | 491 | 37 | 18.33 | 0.0019 | 0.0005 | 5.3E-05 | 0.0032 | 0.0541 | 1 | 0.2168 |
| 0006099 | Tricarboxylic acid cycle | 4 | 79 | 12 | 2.95 | 0.0019 | 0.5303 | 0.0016 | 2.4E-15 | 5.4E-05 | 0.1643 | 1 |
| 0016999 | Antibiotic metabolic process | 5 | 79 | 12 | 2.95 | 0.0019 | 0.5303 | 0.0016 | 2.4E-15 | 5.4E-05 | 0.1643 | 1 |
| 0045333 | Cellular respiration | 5 | 79 | 12 | 2.95 | 0.0019 | 0.5303 | 0.0016 | 2.4E-15 | 5.4E-05 | 0.1643 | 1 |
| 0016053 | Organic acid biosynthetic process | 5 | 214 | 21 | 7.99 | 0.0025 | 0.0540 | 2.1E-05 | 1.4E-09 | 0.3166 | 0.0002 | 0.3585 |
| 0006091 | Generation of precursor metabolites and energy | 4 | 216 | 21 | 8.06 | 0.0025 | 0.4714 | 1.6E-06 | 9.9E-17 | 0.0130 | 0.0155 | 1 |
| 0055085 | Transmembrane transport | 5 | 1,013 | 62 | 37.81 | 0.0025 | 0.1064 | 0.0028 | 0.0719 | 0.3262 | 0.0007 | 0.0040 |
| 0048878 | Chemical homeostasis | 5 | 59 | 10 | 2.2 | 0.0025 | 0.0172 | 0.0177 | 0.1444 | 0.1345 | 0.0077 | 0.6950 |
| 0042592 | Homeostatic process | 4 | 144 | 16 | 5.38 | 0.0039 | 0.0043 | 0.00562 | 0.2749 | 0.0115 | 0.1651 | 1 |
| 0044283 | Small molecule biosynthetic process | 4 | 321 | 26 | 11.98 | 0.0070 | 0.5475 | 1.3E-05 | 3.2E-10 | 0.0197 | 0.0011 | 0.9651 |
| 0055082 | Cellular chemical homeostasis | 4 | 33 | 7 | 1.23 | 0.0072 | 0.0385 | 0.0817 | 0.0556 | 0.3495 | 0.0096 | 0.1560 |
| 0009063 | Cellular amino acid catabolic process | 5 | 9 | 4 | 0.34 | 0.0074 | 0.0515 | 0.0126 | 0.0018 | 0.1313 | 1 | 1 |
| 0016054 | Organic acid catabolic process | 5 | 9 | 4 | 0.34 | 0.0074 | 0.0515 | 0.0126 | 0.0018 | 0.1313 | 1 | 1 |
| 0072524 | Pyridine-containing compound metabolic process | 5 | 138 | 15 | 5.15 | 0.0074 | 0.2710 | 1.6E-06 | 5.9E-09 | 0.0063 | 0.0002 | 0.0971 |
| 0051053 | Negative regulation of DNA metabolic process | 5 | 9 | 4 | 0.34 | 0.0074 | 1 | 0.0126 | 1 | 0.1313 | 0.0656 | 1 |

Annotated: number of genes belonging to the GO terms; Assigned: number of DEGs belongs to the GO terms; Expected: expected number of DEGs for the GO terms if there is no enrichment. The apices of cultivar names were used.

the ripening process. Thus, meta-analysis can be used for gene identification by increasing statistical significance with a greater number of samples.

We further performed motif search for the whole meta-DEGs and for the 484 newly identified DEGs to find possible conserved transcriptional regulation for the meta-DEGs and whether the conserved motifs are shared in the newly identified DEGs or not. The meta-DEGs shared many motifs found in ERF genes and ABA related genes, suggesting hormone-responsive transcriptional regulations are largely controlling transcriptome changes in strawberry

**Table 3. Enriched GO terms for down-regulated meta-DEGs and FDR values for down-regulated DEGs (FR vs. BG) in the six cultivars.**

| GO id | Term | GO level | Anno. | Assig. | Exp. | FDR | | | | | | |
|---|---|---|---|---|---|---|---|---|---|---|---|---|
| | | | | | | Meta | Toyo | Beni | Xiao | Snow | Sunny | Kings |
| 0015979 | Photosynthesis | 4 | 62 | 34 | 7.11 | 1.6E-13 | 2.2E-08 | 2.1E-06 | 4.9E-11 | 2.5E-09 | 9.4E-20 | 1.1E-24 |
| 0042546 | Cell wall biogenesis | 4 | 136 | 49 | 15.6 | 3.5E-11 | 2.9E-08 | 4.1E-16 | 1.3E-13 | 1.1E-08 | 2.9E-09 | 4.4E-07 |
| 0010383 | Cell wall polysaccharide metabolic process | 5 | 90 | 38 | 10.32 | 3.9E-11 | 5.1E-09 | 1.5E-16 | 5.8E-12 | 2.1E-09 | 1.1E-10 | 2.5E-09 |
| 0044264 | Cellular polysaccharide metabolic process | 5 | 135 | 47 | 15.48 | 1.8E-10 | 3.6E-12 | 1.1E-15 | 1.9E-09 | 1.2E-15 | 1.5E-11 | 4.1E-10 |
| 0005976 | Polysaccharide metabolic process | 5 | 136 | 47 | 15.6 | 1.8E-10 | 3.6E-12 | 1.1E-15 | 2.2E-09 | 1.2E-15 | 1.5E-11 | 4.1E-10 |
| 0044262 | Cellular carbohydrate metabolic process | 4 | 136 | 47 | 15.6 | 1.8E-10 | 3.6E-12 | 1.1E-15 | 6.3E-10 | 1.2E-15 | 1.5E-11 | 4.1E-10 |
| 0044038 | Cell wall macromolecule biosynthetic process | 5 | 30 | 20 | 3.44 | 1.8E-10 | 1.2E-08 | 1.2E-11 | 3.4E-10 | 5.5E-08 | 1.5E-11 | 8.2E-08 |
| 0070589 | Cellular component macromolecule biosynthetic process | 4 | 30 | 20 | 3.44 | 1.8E-10 | 1.2E-08 | 1.2E-11 | 3.4E-10 | 5.5E-08 | 1.5E-11 | 8.2E-08 |
| 0044036 | Cell wall macromolecule metabolic process | 4 | 126 | 43 | 14.45 | 1.5E-09 | 1.9E-08 | 3.7E-13 | 1.6E-11 | 7.4E-09 | 2.0E-08 | 4.1E-07 |
| 0034637 | Cellular carbohydrate biosynthetic process | 5 | 62 | 26 | 7.11 | 8.5E-08 | 1.6E-09 | 1.6E-08 | 4.1E-08 | 2.0E-13 | 8.6E-12 | 2.5E-09 |
| 0016051 | Carbohydrate biosynthetic process | 5 | 87 | 30 | 9.98 | 9.9E-07 | 1.3E-06 | 6.1E-08 | 1.8E-08 | 1.5E-10 | 2.1E-10 | 1.8E-07 |
| 0010154 | Fruit development | 5 | 17 | 12 | 1.95 | 1.3E-06 | 0.4582 | 3.6E-09 | 1.8E-08 | 0.0010 | 0.9913 | 0.0052 |
| 0048316 | Seed development | 4 | 17 | 12 | 1.95 | 1.3E-06 | 0.4582 | 3.6E-09 | 1.8E-08 | 0.0010 | 0.9913 | 0.0052 |
| 0019684 | Photosynthesis, light reaction | 5 | 32 | 16 | 3.67 | 5.3E-06 | 0.0246 | 0.1177 | 0.0070 | 0.0530 | 8.3E-10 | 2.0E-11 |
| 0010087 | Phloem or xylem histogenesis | 5 | 39 | 16 | 4.47 | 0.0001 | 0.0197 | 0.0006 | 0.0008 | 0.0015 | 4.5E-07 | 3.7E-09 |
| 0009888 | Tissue development | 4 | 83 | 24 | 9.52 | 0.0007 | 0.0280 | 0.0097 | 0.1880 | 0.0185 | 0.0011 | 0.0003 |
| 0005975 | Carbohydrate metabolic process | 4 | 391 | 73 | 44.84 | 0.0007 | 0.00115 | 4.7E-06 | 0.0010 | 2.4E-06 | 2.3E-05 | 9.7E-06 |
| 0007018 | Microtubule-based movement | 4 | 226 | 48 | 25.92 | 0.0008 | 3.8E-09 | 0.0034 | 0.3796 | 1.0E-30 | 0.1102 | 0.4761 |
| 0010109 | Regulation of photosynthesis | 5 | 5 | 5 | 0.57 | 0.0009 | 0.0382 | 0.0188 | 0.0122 | 0.0332 | 0.0087 | 0.0016 |
| 0043467 | Regulation of generation of precursor metabolites, energy | 5 | 5 | 5 | 0.57 | 0.0009 | 0.0382 | 0.0188 | 0.0122 | 0.0332 | 0.0087 | 0.0016 |
| 0009415 | Response to water | 4 | 32 | 13 | 3.67 | 0.0010 | 0.4469 | 3.9E-05 | 6.7E-05 | 0.0019 | 0.0005 | 2.0E-05 |
| 0001101 | Response to acid chemical | 4 | 140 | 33 | 16.05 | 0.0015 | 0.0821 | 0.0001 | 0.0005 | 4.7E-05 | 0.1692 | 0.3296 |
| 0006662 | Glycerol ether metabolic process | 5 | 86 | 23 | 9.86 | 0.0027 | 0.0010 | 0.2188 | 0.0354 | 1 | 0.0006 | 2.0E-05 |
| 0018904 | Ether metabolic process | 4 | 86 | 23 | 9.86 | 0.0027 | 0.0010 | 0.2188 | 0.0354 | 1 | 0.0006 | 2.0E-05 |
| 0031122 | Cytoplasmic microtubule organization | 5 | 12 | 7 | 1.38 | 0.0040 | 0.0626 | 0.0258 | 0.2315 | 0.0530 | 0.1580 | 1 |
| 0003002 | Regionalization | 4 | 4 | 4 | 0.46 | 0.0051 | 0.0130 | 0.0054 | 1 | 0.0091 | 1 | 1 |
| 0009956 | Radial pattern formation | 5 | 4 | 4 | 0.46 | 0.0051 | 0.0130 | 0.0054 | 1 | 0.0091 | 1 | 1 |
| 0015995 | Chlorophyll biosynthetic process | 5 | 4 | 4 | 0.46 | 0.0051 | 0.0130 | 0.0054 | 0.0039 | 0.0091 | 0.0027 | 0.1843 |
| 1901700 | Response to oxygen-containing compound | 4 | 151 | 33 | 17.32 | 0.0053 | 0.1244 | 0.0007 | 0.0017 | 0.0001 | 0.2955 | 0.4995 |

*(Continued)*

**Table 3.** (Continued)

| GO id | Term | GO level | Anno. | Assig. | Exp. | FDR | | | | | | |
|-------|------|----------|-------|--------|------|------|------|------|------|------|------|------|
| | | | | | | Meta | Toyo | Beni | Xiao | Snow | Sunny | Kings |
| 0072348 | Sulfur compound transport | 5 | 29 | 11 | 3.33 | 0.0059 | 0.0130 | 0.0065 | 0.0282 | 0.0707 | 0.0562 | 0.0034 |
| 0010035 | Response to inorganic substance | 4 | 112 | 26 | 12.84 | 0.0085 | 0.0246 | 0.0047 | 0.0017 | 0.0001 | 0.2750 | 0.0728 |

Anno., annotated; Assig., Assigned; Exp., expected. The apices of cultivar names were used.

fruit ripening (S1 Fig). Furthermore, the newly identified meta-DEGs also shared such motifs indicating the genes likely have altered expression during strawberry fruit ripening (S1 Fig).

## Changes in anthocyanin biosynthesis-related genes

Coloration of strawberry fruit during ripening is one of the most prevalent and economically important changes and has been intensively investigated. The color component is mainly anthocyanin, a water-soluble flavonoid compound, with pelargonidin being the major anthocyanin accumulated during ripening [32]. Structural and regulatory genes in the anthocyanin biosynthesis pathway have been investigated in many crop species including strawberry. MYB, bHLH, and WD40 repeat proteins are assembled into an MBW complex and control anthocyanin biosynthesis [33].

There are 1,213 MYB-annotated genes, among which 719 were expressed in fruit tissue and 423 were differentially expressed in at least one of the six cultivars. The meta-DEGs included 191 *MYB* genes, 155 down-regulated and 36 up-regulated at FR compared with BG stage. *FaMYB1* (AF401220) and *FaMYB10* (EU155162) are ripening-related transcription factors, known to be negative or positive regulators of anthocyanin biosynthesis [3, 34, 35]. We identified three orthologs for each of the *FaMYB1* and *FaMYB10* genes in the *F. × ananassa* reference genome (https://www.rosaceae.org) with high significance (E-value $< 10^{-8}$) in BLAST searches: FxaC_19g15290, FxaC_20g18010, and FxaC_18g28180 for *FaMYB1* and FxaC_3g25830, FxaC_4g15020, and FxaC_2g30690 for *FaMYB10*, respectively. Two of the *MYB1* orthologs and all of the *MYB10* orthologs were up-regulated meta-DEGs (**Fig 3**). The three *FaMYB10* genes were designated as *FaMYB10-1* (FxaC_4g15020), *FaMYB10-2* (FxaC_2g30690), *FaMYB10-3B* (FxaC_3g25830) and the FaMYB10-2 was recently confirmed to be involved in coloration of strawberry fruit [36].

A total of 113 bHLH orthologs have been identified in the *F. vesca* genome, and their transcription profiles have been investigated in three different-colored cultivars, 'Benihoppe', 'Snow princess', and 'Xiobai' [37]. From this analysis, Zhao *et al.* suggested that seven bHLH genes are involved in anthocyanin biosynthesis [37]. We examined transcript profiles of these seven bHLH genes in three more strawberry cultivars. There are 454 genes annotated as bHLH genes in the *F. × ananassa* reference genome, which is about four times as many as in *F. vesca*, consistent with the octoploid origin of *F. × ananassa*. Among these, 282 genes were expressed (FPKM $> 0.3$) in at least one of the fruit samples and 190 genes were differentially expressed between BG and FR in at least one of the six cultivars. For *bHLH* genes, we identified 97 meta-DEGs comprising 76 up-regulated and 21 down-regulated genes at the FR compared to BG stage. Among the seven *bHLH* genes suggested to be involved in fruit anthocyanin biosynthesis [37], all three orthologs of *FvbHLH27* and one ortholog for *FvbHLH40* were up-regulated. By contrast, one ortholog of the *FvbHLH80* and three *FvbHLH98* orthologs were down-regulated (**Fig 3A**).

The other component of the MBW complex is WD40 repeat-containing proteins. There were 1,028 genes annotated as WD40 repeat-containing proteins of which 137 were not

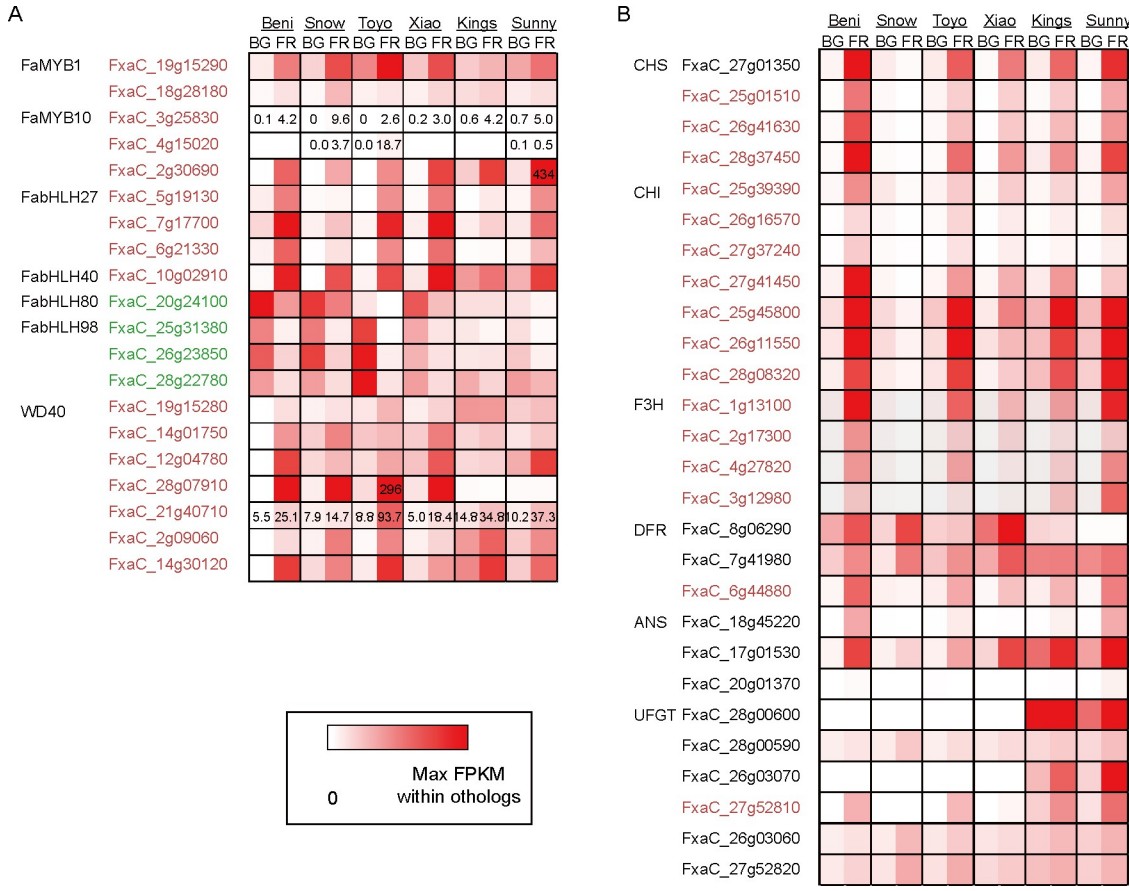

**Fig 3.** Heatmap of anthocyanin biosynthesis regulatory genes (A) and structural genes (B). Up- and down-regulated meta-DEGs between big green (BG) and fully red (FR) stages are colored red and green, respectively. Average FPKM values for developmental stages of the six cultivars are color-scaled the same for each gene ortholog. The numbers for FPKM values were presented if necessary. CHS: chalcone synthase; CHI: chalcone isomerase; F3H: flavanone3 hydroxylase; DFR: dihydroflavonol 4-reductase; ANS: anthocyanidin synthase; UFGT: UDP-glucose:flavonoid 3-O-glycosyltransferase.

expressed in samples from any of the six cultivars and 693 did not show any changes in expression between the BG and FR stages in the six cultivars. We identified 60 meta-DEGs: 33 genes up-regulated and 27 down-regulated in FR compared with BG samples. Since there have been few studies on these WD40 repeat genes, whose expressions corresponded to other MBW complex genes were selected as possible candidates that were up-regulated in most of the cultivars. From the expression analysis of the six cultivars and meta-analysis, we propose that the WD40 genes listed in **Fig 3**, especially FxaC_21g40710, which was significantly up-regulated in all cultivars except 'Snow princess', could be components of an MBW complex with FaMYB1 and FabHLH27 or FabHLH40. If this is the case, the *WD40* gene could be also critical for white flesh color in the cultivar 'Snow princess' along with FaMYB10 [38]. Further analysis and experimental evidences would be necessary for testing this possibility.

We also observed changes in expression of anthocyanin biosynthesis structural genes encoding enzymes such as chalcone synthase (CHS), chalcone isomerase (CHI), flavanone 3-hydroxylase (F3H), dihydroflavonol 4-reductase (DFR), anthocyanidin synthase (ANS), and UDP-glucose: flavonoid 3-O-glycosyltransferase (UFGT) (**Fig 3B**). Clear differences between white- and red-fruit cultivars were observed in genes encoding the first three enzymes, CHS, CHI, and F3H. Orthologs of these genes were up-regulated in all red cultivars at the FR stage

compared to BG but down-regulated in 'Snow princess'. 'Xiaobai', which has red skin but white flesh, showed a relatively smaller increase in expression of genes encoding CHI and F3H at the FR stage than the red-skin and red-flesh cultivars. The similar response of these three structural genes supports the idea that a regulatory gene is the determinant of white-color fruit cultivars and this could be a *WD40* gene, a component of the MYB-bHLH-WD40 (MBW) complex.

## Other aspects of strawberry ripening

MADS box genes are related to many aspects of plant development as well as fruit ripening [39]. There are 255 MADS-annotated genes in the strawberry reference genome and 97 of them were expressed in samples of at least one stage from the six cultivars with FPKM value higher than 0.3. Among these, 54 genes were differentially expressed in at least one of the six cultivars and 34 genes were selected as meta-DEGs with 26 down-regulated and eight up-regulated at FR compared with BG stage. However, FxaC_13g22210, corresponding to *FaMADS1* (GQ398009) involved in strawberry fruit ripening, was up-regulated in 'Snow princess' and 'Toyonoka' only [40]. *FaMADS9* ortholog FxaC_22g08610 was also up-regulated in 'Snow princess' and 'Toyonoka' only [39]. These results suggest that MADS box genes are more likely involved in fruit development than ripening and support the idea that different cultivars exhibit inconsistency in their development and coloration.

Min *et al*. suggest that several transcription factors are involved in fruit ripening and possibly related to postharvest storability [30]. These candidates are NAC83, WRKY40, and WRKY48, encoded by genes corresponding to FxaC_13g22700, FxaC_24g33610, and FxaC_14g18300 in the new strawberry reference genome. We also determined these to be meta-DEGs, supporting the possibility that these genes are involved in ripening and postharvest storability in a wide range of cultivars.

A relationship between ripening and ubiquitin-mediated proteolysis was proposed and investigated [29]. A number of ubiquitination-related genes were also included among the meta-DEGs, such as ubiquitin-activating enzyme (E1), ubiquitin-conjugating enzyme (E2), and ubiquitin-protein ligase (E3). Among the 21, 203, and 359 genes annotated as E1, E2, and E3, respectively, we selected 0, 14, and 36 as meta-DEGs. These results support the relationship between ripening and ubiquitination as previously suggested [29].

## Discussion

We performed a meta-analysis to obtain an in-depth view of strawberry ripening using publicly available strawberry RNA-Seq data. We mapped raw reads of previous data to the new strawberry reference genome [1] and elucidated DEGs in six cultivars as well as meta-DEGs.

Genetic relationships among the six cultivars and reference cultivar 'Camarosa' were investigated from SNP genotypes based on genic sequences. Due to the high level of polyploidy and heterozygotic nature of asexually propagated strawberry cultivars, a limited number of loci could be genotyped; however, to the best of our knowledge, this is the largest genome-wide genotype comparison among *F.* × *ananassa* cultivars. Since strawberry breeding history is relatively short and cultivar diversity is limited, this information will be valuable for planning breeding programs and designing future experiments with different strawberry cultivars [41].

Enriched GO terms associated with the DEGs showed consistent results with previous studies but also revealed the characteristics of the six cultivars. Notably, enriched GO terms associated with down-regulated DEGs were more consistent among meta-DEGs and the six cultivars than those of up-regulated DEGs. Moreover, the meta-DEGs allowed us an expanded view of changes at different ripening stages, which could not be discovered from a single

study. We additionally identified 483 meta-DEGs not revealed in any of the single-cultivar studies along with 695 meta-DEGs detected in only one of the six cultivars. This was possible by increasing the statistical significance of the meta-analysis with an increased number of cultivars.

We selected fruit coloration as a comparable phenotype in the six cultivars and showed that meta-analysis could empower investigation of the underlying mechanism. From the meta-analysis we were able to find a candidate gene (FxaC_21g40710) for a WD40 protein, which possibly forms an MBW complex with FaMYB1 and FabHLH27 or FabHLH40. We also propose that *WD40* genes could be color-determinant genes in the six strawberry cultivars. Further investigations of this *WD40* candidate should be performed.

Color can be determined qualitatively and is subject to little environmental effect; thus, it is appropriate for the combined analysis of cultivars grown under different conditions. Phenotyping applied together or a standard phenotype index for the six cultivars would make the meta-DEGs more powerful for understanding the transcriptomic contribution to the phenotype. Furthermore, standard gene ID and accumulated expression data should be combined with a user-friendly interface for future strawberry research.

In summary, our meta-analysis of public transcriptome data provides an expanded view of strawberry ripening including common and cultivar-specific transcriptome changes. This work demonstrates that meta-analysis of existing transcriptome data can provide a deep understanding of specific processes not revealed by a single study. As the number of transcriptome studies increases and data accumulate, systematic analysis using meta-analysis will be necessary for maximizing data utility and addressing biological questions.

## Supporting information

**S1 Checklist.**
(DOCX)

**S1 Fig. Significantly enriched motifs in promoters of meta-DEGs.** Enriched motifs sequences were searched in all meta-DEGs (12,339 genes) (A) and newly identified meta-DEGs (483 genes) (B).
(TIF)

**S1 Table. RNA-Seq reads and mapping summary for studies used in this analysis.**
(DOCX)

**S2 Table. Pearson's correlation coefficient ($R^2$) between samples based on their expression profiles.**
(DOCX)

**S3 Table. Number of DEGs in each study and DEGs retained after meta-analysis.**
(DOCX)

**S4 Table. The 483 meta-DEGs from meta-analysis of the six cultivars.**
(XLSX)

## Acknowledgments

The authors thank Jong Seung Kim MD, PhD (Chonbuk National University Hospital) for his technical advice for meta-analysis.

## Author Contributions

**Conceptualization:** Gibum Yi, Eun Jin Lee.

**Data curation:** Hosub Shin.

**Investigation:** Gibum Yi, Hosub Shin, Kyeonglim Min.

**Supervision:** Eun Jin Lee.

**Writing – original draft:** Gibum Yi.

**Writing – review & editing:** Gibum Yi, Eun Jin Lee.

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
