## [Decision Letter · Decision Letter 0]

23 Feb 2021

PONE-D-20-40314

Comprehensive transcriptomic view of strawberry fruit ripening through meta-analysis

PLOS ONE

Dear Dr. Yi,

Thank you for submitting your manuscript to PLOS ONE. After careful consideration, we feel that it has merit but does not fully meet PLOS ONE’s publication criteria as it currently stands. Therefore, we invite you to submit a revised version of the manuscript that addresses the points raised during the review process.

The ms «Comprehensive transcriptomic view of strawberry fruit ripening through meta-analysis» was reviewed by three specialists who addressed different comments that can assist the authors in preparing a new version.

It is confirmed that the metaanalysis approach represents an advantage as compared to experiments with few biological replicates.

However, considering the main objective of the research that is to spot «the genetic and physiological differences among cultivars that preclude consensus understanding of the ripening processes», the number of cultivars (6) and two phases (2)  is limited. So the objective might be adapted to the setup of the metaanalysis.

Also the proposal advanced for a candidate gene might be supported by a more grounded research.

We look forward to receiving your revised manuscript.

Kind regards,

Sara Amancio

Academic Editor

PLOS ONE

Journal Requirements:

3. We note that this manuscript is a systematic review or meta-analysis; our author guidelines therefore require that you use PRISMA guidance to help improve reporting quality of this type of study. Please upload copies of the completed PRISMA checklist as Supporting Information with a file name “PRISMA checklist”.

Reviewers' comments:

Reviewer's Responses to Questions

**Comments to the Author**

1. Is the manuscript technically sound, and do the data support the conclusions?

Reviewer #1: Yes

Reviewer #2: Yes

Reviewer #3: Partly

2. Has the statistical analysis been performed appropriately and rigorously? 

Reviewer #1: I Don't Know

Reviewer #2: Yes

Reviewer #3: Yes

3. Have the authors made all data underlying the findings in their manuscript fully available?

Reviewer #1: Yes

Reviewer #2: Yes

Reviewer #3: Yes

4. Is the manuscript presented in an intelligible fashion and written in standard English?

Reviewer #1: Yes

Reviewer #2: Yes

Reviewer #3: Yes

5. Review Comments to the Author

Reviewer #1: You say that genetic and physiological variation among cultivars preclude an “consensus” understanding of these processes (line 20-21), but you then go on to say that you’ve provided a comprehensive view of strawberry ripening with only 6 cultivars (line 24). These statements are at odds. If variation among cultivars/genotypes is one of the major challenges, it seems doubtful that 6 cultivars can provide a comprehensive view.

Line 58-59: I found the statement “Understanding the genetic relationships among current cultivars is therefore necessary for synergistic analysis.” a bit vague. What are these synergistic analyses?

Line 96: You need to define the acronyms ‘BG’ and ‘FR’ the first time you use them in the paper

Line 113: How do we know that “Large Green” in one project means the same thing as “Big Green” or “Mature Green” in another project? Can you compare how these experiments defined these developmental stages?

Line 113: Again, you can’t really call this meta-analysis comprehensive, with only 2 tissue stages and 6 genotypes. I think it is more accurate to call it a very deep dive into comparing the large green and large red fruit stages. A LOT goes on between those stages, e.g. during the large white, and pink/turning stages, when many genes are being activated.

Line 156: in the statement “genetic distance of these two cultivars was very close”, the word close should be replaced with “small” or “low”.

Line 162-163: While strawberry cultivars are individual diverse, with only 6 cultivars and most coming from Asia, this statement is almost certainly false.

Line 271: This section focuses on transcription factors impacting anthocyanin biosynthesis. There was a 2020 study published in The Plant Cell that identifies the MYB gene(s) controlling white vs. red fruit trait in octoploid strawberry. This study should have been discussed, and you should have compared your meta-DEGs with the genes they functionally characterized.

Figure 2: Panels B and C are indecipherable.

Reviewer #2: The manuscript focuses its attention on meta-analysis of NGS transcriptomic data due to the release of the new strawberry reference genome. Using the power of meta-analysis and combining chosen and comparable datasets the author provided an overview of the transcriptomic change during fruit ripening (or at least differences between unripe and ripe stage) adding new DEG impossible to find in a single experiment with few biological replicates.

Below some suggestion that refer to row number of the manuscript pdf file

88-89 remove repetition of independent

96 Add Full Red (FR) and Big Green (BG) to their acronym and in table 1 please add next to developmental stages with name different than the one you used the corresponding acronym i.e. large green (BG) in order to make easier table interpretation

102 add edgeR package version to allow reproducibility

110 add FactoMineR package version to allow reproducibility

118 add TopGo package version to allow reproducibility

156-157 their transcript profiles at the BG stage were distinct enough to consider them different cultivars… can you give a numerical or statistical interpretation of this “distinct anough”?

181 word genes contain a number 1… I think it’s bibliography?

192 as above study has a 27 in apex position

200 consider changing "specific in two cultivars" into "two-cultivar-specific"… I think it’s easier to understand and follow your statement

221 GO terms associated with fruit and seed development were enriched in four of the six cultivars, suggesting variation in developmental timing (Tables 2 and 3). ‘Toyonoka’ and Sunnyberry ‘Sunnyberry’ might develop earlier than the other cultivars and have finished seed development by the BG stage… it could be true for seed but it is difficult for fruit because otherwise fruit ripening should also different from color changing! Couldn’t be just fruit harvested earlier than the other cultivars?

230-233 this sentence is a bit hard to interpret (very full of information) … can you rephrase a bit?

247-255 it’s partly a repetition of what said just before (221-230). I suggest to merge them in only one of the two section

274 pelaRgonidin

288 all of the MYB10 orthologs were up-regulated meta-DEGs (Fig 3)… it’s not easy to draw this conclusion looking only at the figure… can’t you make it two color scale or add numbers in some way? Else is necessary to look at raw data!

307-309 There is no FvbHLH7 in the picture. Anyway I can’t really understand what you mean in this sentence.

318-320 FxaC_21g40710, which was significantly up-regulated in all cultivars except ‘Snow princess’… again is difficult to appreciate this fact from the picture

321 FaMYB10-2 right?

318-321 This is a bit speculative, but I like it in a paper like this!

329 white flesh red skin right?

374 what do you mean by “consistent”? Can you explain it better?

380 (We selected fruit coloration as a comparable phenotype ) and subsequently row 386 … this statement is in contrast with what you say at row 221 that is Toyonoka’ and ‘Sunnyberry’ might develop earlier than the other cultivars and have finished seed development by the BG stage. If that is the case, fruit coloration isn’t anymore a comparable phenotype!

Reviewer #3: - Using a meta-analysis approach the Authors state to have been able to find a candidate gene (FxaC_21g40710) for a WD40 protein, which *possibly* forms a complex with FaMYB1 and FabHLH27 or FabHLH40. They also propose that WD40 genes could be color-determinant genes in the six strawberry cultivars, yet asserting that further investigations of this WD40 candidate should be performed.

In my opinion, a more conclusive molecular evidence should be provided to confirm such a statement and the actual role of the candidate gene.

- DEG promoter motif analysis should be conducted to test possible enrichment of putative regulatory cis-acting elements. See for example the paper of Shaar-Moshe et al. (2015) cited by the Authors. If not possible in strawberry, Authors should clearly explain why it was not performed.

- Table 2 summarizes the number of genes identified belonging to the given GO terms and lists the striking number of 1,013 genes falling into the category “Transmembrane transport”. This fact should be emphasized and properly addressed in the discussion.

In the same Table, the most significant FDR values should be highlighted in bold or some other way.

---List of corrections and typos to address---

- Cultivar names should be written in apices (e.g. ‘Toyo’) throughout the text, including tables and figures.

- The ‘MBW’ acronym is reported four times in the manuscript without any specification of its meaning.

- Table 1: The header row should be written in bold to facilitate the reader.

- Table 1: The header of the first column of Table 1 should be changed into “Project number” and, in the table legend, the database source (GenBank?) should be specified.

- Table 1: The header of the fourth column (“Reps”) should be clearly defined and explained in the legend.

- Table 1: Correct “HiSeq x ten” by “HiSeq x Ten”

- Table 1: The header of the sixth column “( x 1000 ea)” should be corrected to “(x 1,000)”.

- Table 1. All the three samples of PRJNA552213 should be marked as “NA” in the “Reference” column.

- Line 235: correct “belongs” by “belonging”

- Line 274: correct “pelagonidin” by “pelargonidin”

6. PLOS authors have the option to publish the peer review history of their article (what does this mean?). If published, this will include your full peer review and any attached files.

Reviewer #1: No

Reviewer #2: **Yes: **Remo Chiozzotto

Reviewer #3: No

---

## [Author Response · Author response to Decision Letter 0]

12 Apr 2021

5. Review Comments to the Author

Reviewer #1: You say that genetic and physiological variation among cultivars preclude an “consensus” understanding of these processes (line 20-21), but you then go on to say that you’ve provided a comprehensive view of strawberry ripening with only 6 cultivars (line 24). These statements are at odds. If variation among cultivars/genotypes is one of the major challenges, it seems doubtful that 6 cultivars can provide a comprehensive view.

Thanks for your critical comment. We basically agree with your point. Now we changed the word ‘comprehensive’ to ‘expanded’ or ‘in-depth’ throughout the manuscript.

Line 58-59: I found the statement “Understanding the genetic relationships among current cultivars is therefore necessary for synergistic analysis.” a bit vague. What are these synergistic analyses?

The word 'synergistic’ is changed to 'combined’ for clarity.

Line 96: You need to define the acronyms ‘BG’ and ‘FR’ the first time you use them in the paper

Thanks for your comment. The acronyms were provided in the method section.

Line 113: How do we know that “Large Green” in one project means the same thing as “Big Green” or “Mature Green” in another project? Can you compare how these experiments defined these developmental stages?

The data set from Min et al. (PRJNA564159) and Hu et al. (PRJNA394190) both referred to the same reference ‘Fait et al. 2008 Plant Physiol.’ for their sampling stages. The other public data set (PRJNA552213) which is not published in a paper is little vague but the PCA analysis showed consistencies for their expression profiles.

Line 113: Again, you can’t really call this meta-analysis comprehensive, with only 2 tissue stages and 6 genotypes. I think it is more accurate to call it a very deep dive into comparing the large green and large red fruit stages. A LOT goes on between those stages, e.g. during the large white, and pink/turning stages, when many genes are being activated.

Thanks for your comment. 'Expanded’ or 'in-depth’ look more accurate for this study.

Line 156: in the statement “genetic distance of these two cultivars was very close”, the word close should be replaced with “small” or “low”.

Thanks for the indication. The word ‘close’ was replace to ‘small’.

Line 162-163: While strawberry cultivars are individual diverse, with only 6 cultivars and most coming from Asia, this statement is almost certainly false.

The sentence was revised as ‘These cultivars were genetically diverse to cover certain amount of genetic diversities of strawberry’. Please see lines 172-173 in the Revised Manuscript with Track Changes.

Line 271: This section focuses on transcription factors impacting anthocyanin biosynthesis. There was a 2020 study published in The Plant Cell that identifies the MYB gene(s) controlling white vs. red fruit trait in octoploid strawberry. This study should have been discussed, and you should have compared your meta-DEGs with the genes they functionally characterized.

Thanks for your comment. The study was now referred in the Results (Lines 306-308).

Figure 2: Panels B and C are indecipherable.

The meaning of the numbers for panels B and C is explained in the figure legend.

Reviewer #2: The manuscript focuses its attention on meta-analysis of NGS transcriptomic data due to the release of the new strawberry reference genome. Using the power of meta-analysis and combining chosen and comparable datasets the author provided an overview of the transcriptomic change during fruit ripening (or at least differences between unripe and ripe stage) adding new DEG impossible to find in a single experiment with few biological replicates.

Below some suggestion that refer to row number of the manuscript pdf file

88-89 remove repetition of independent

Thanks for the indication. The repetition was removed.

96 Add Full Red (FR) and Big Green (BG) to their acronym and in table 1 please add next to developmental stages with name different than the one you used the corresponding acronym i.e. large green (BG) in order to make easier table interpretation

Thanks for your comment. The acronym was added in the method section and only acronym was used in the following sections.

102 add edgeR package version to allow reproducibility

Thanks, the version was added.

110 add FactoMineR package version to allow reproducibility

Thanks, the version was added.

118 add TopGo package version to allow reproducibility

Thanks, the version was added.

156-157 their transcript profiles at the BG stage were distinct enough to consider them different cultivars… can you give a numerical or statistical interpretation of this “distinct anough”?

Well, it’s hard to give numerical interpretation but from the PCA the distance between Xiaobai and Bennihoppe in BG stages is farther than Bennihoppe and Snow Princess.

181 word genes contain a number 1… I think it’s bibliography?

Thanks for your kind indication. We changed it to a correct form.

192 as above study has a 27 in apex position

We also changed it to a correct form.

200 consider changing "specific in two cultivars" into "two-cultivar-specific"… I think it’s easier to understand and follow your statement

Thanks, we changed it as recommend.

221 GO terms associated with fruit and seed development were enriched in four of the six cultivars, suggesting variation in developmental timing (Tables 2 and 3). ‘Toyonoka’ and Sunnyberry ‘Sunnyberry’ might develop earlier than the other cultivars and have finished seed development by the BG stage… it could be true for seed but it is difficult for fruit because otherwise fruit ripening should also different from color changing! Couldn’t be just fruit harvested earlier than the other cultivars?

Thanks for your helpful comment. The possibility we suggested is quite hasty since the GO term only have 17 annotated and 12 assigned genes. We also agree that the different set of samples from independent studies could affect the transcriptomic differences. Thus we decided to remove the sentences (L232-237).

230-233 this sentence is a bit hard to interpret (very full of information) … can you rephrase a bit?

Thanks for your indication. We rephrased the sentence. Please see lines 241-244.

247-255 it’s partly a repetition of what said just before (221-230). I suggest to merge them in only one of the two section

Thanks for your suggestion. But we explained the characteristics of the meta-DEGs in (247-255) whereas, lines 221-230 is more about comparisons among the six cultivars. 

274 pelaRgonidin

Thanks. Corrected.

288 all of the MYB10 orthologs were up-regulated meta-DEGs (Fig 3)… it’s not easy to draw this conclusion looking only at the figure… can’t you make it two color scale or add numbers in some way? Else is necessary to look at raw data!

Thanks for your suggestion. We now add numbers for FPKM values if necessary in Fig 3.

307-309 There is no FvbHLH7 in the picture. Anyway I can’t really understand what you mean in this sentence.

Thanks for your indication. From the previous study by Zhao et al. (2018 Sci Rep), seven Fragaria vesca bHLH (FvbHLH) genes were selected for their involvement in fruit anthocyanin biosynthesis. We searched orthologs for these genes and showed their expression when the orthologs are in the meta-DEGs.

318-320 FxaC_21g40710, which was significantly up-regulated in all cultivars except ‘Snow princess’… again is difficult to appreciate this fact from the picture

Fig 3. was polished to clearly show the expression differences by adding FPKM values.

321 FaMYB10-2 right?

That’s right. Wang et al.(2020 Plant Biotechnol J) used FaMYB10-2 as a name of allele for FaMYB10 gene. In another paper (Castillejo et al. 2020 Plant Cell) which we add as a reference in the revised manuscript used FaMYB10-1, FaMYB10-2 for orthologous genes of FaMYB10. We don’t want to make any confusion regarding the FaMYB10-2 in this manuscript.

318-321 This is a bit speculative, but I like it in a paper like this!

We agree with you. We add a sentence to ‘Further analysis and experimental evidences will be necessary for testing this possibility’. Please see line 344 (file with track changes).

329 white flesh red skin right?

You are right. We revised it.

374 what do you mean by “consistent”? Can you explain it better?

Thanks for your comment. We specified the sentence in lines 398-399.

380 (We selected fruit coloration as a comparable phenotype ) and subsequently row 386 … this statement is in contrast with what you say at row 221 that is Toyonoka’ and ‘Sunnyberry’ might develop earlier than the other cultivars and have finished seed development by the BG stage. If that is the case, fruit coloration isn’t anymore a comparable phenotype!

That’s right. We removed the sentences in lines 232-237.

Reviewer #3: - Using a meta-analysis approach the Authors state to have been able to find a candidate gene (FxaC_21g40710) for a WD40 protein, which *possibly* forms a complex with FaMYB1 and FabHLH27 or FabHLH40. They also propose that WD40 genes could be color-determinant genes in the six strawberry cultivars, yet asserting that further investigations of this WD40 candidate should be performed.

In my opinion, a more conclusive molecular evidence should be provided to confirm such a statement and the actual role of the candidate gene.

Thanks for your comments. We agree with your comment. We only suggest the possibility and also mentioned the requirement of further molecular evidences as your comment.

- DEG promoter motif analysis should be conducted to test possible enrichment of putative regulatory cis-acting elements. See for example the paper of Shaar-Moshe et al. (2015) cited by the Authors. If not possible in strawberry, Authors should clearly explain why it was not performed.

Thanks for your recommendation. We further performed motif analysis and it was described in method section (135-141) and in the results (281-287)

- Table 2 summarizes the number of genes identified belonging to the given GO terms and lists the striking number of 1,013 genes falling into the category “Transmembrane transport”. This fact should be emphasized and properly addressed in the discussion.

In the same Table, the most significant FDR values should be highlighted in bold or some other way.

The number of assigned gene does not tell about the fruit samples we used. The numbers are just assigned to the GO term from the whole annotated genes for the reference genome.

--List of corrections and typos to address---

- Cultivar names should be written in apices (e.g. ‘Toyo’) throughout the text, including tables and figures.

We only used apices in Table 2 and 3, in the text we think full name is more appropriate as the references.

- The ‘MBW’ acronym is reported four times in the manuscript without any specification of its meaning.

The full name of ‘MBW’ was provided at the first appearance.

- Table 1: The header row should be written in bold to facilitate the reader.

Thanks. Changed.

- Table 1: The header of the first column of Table 1 should be changed into “Project number” and, in the table legend, the database source (GenBank?) should be specified.

The header was changed and specified in the legend.

- Table 1: The header of the fourth column (“Reps”) should be clearly defined and explained in the legend.

Thanks. Explained in the legend.

- Table 1: Correct “HiSeq x ten” by “HiSeq x Ten”

Thanks. Corrected.

- Table 1: The header of the sixth column “( x 1000 ea)” should be corrected to “(x 1,000)”.

Thanks. Corrected.

- Table 1. All the three samples of PRJNA552213 should be marked as “NA” in the “Reference” column.

Thanks. Corrected.

- Line 235: correct “belongs” by “belonging”

Thanks. Corrected.

- Line 274: correct “pelagonidin” by “pelargonidin”

Thanks. Corrected.

---

## [Decision Letter · Decision Letter 1]

26 Apr 2021

PONE-D-20-40314R1

Expanded transcriptomic view of strawberry fruit ripening through meta-analysis

PLOS ONE

Dear Dr. Yi,

Thank you for submitting your manuscript to PLOS ONE. After careful consideration, we feel that it has merit but does not fully meet PLOS ONE’s publication criteria as it currently stands. Therefore, we invite you to submit a revised version of the manuscript that addresses the points raised during the review process.

You are almost there.

Please take into account the comments by the reviewer:

1) FxaC_4g15020 is referred as FaMYB10-1 in ref n. 36 (Allelic Variation of MYB10 Is the Major Force Controlling Natural Variation in Skin and Flesh Color in Strawberry (Fragaria spp.) Fruit)

2) Chromosomes Fvb1-1 and Fvb1-2 carry one FaMYB10 homoeolog each: FaMYB10-1 (maker-Fvb1-1-snap-gene-139.18 or FxaC_4g15020) and FaMYB10-2 (maker-Fvb1-2-snap-gene-157.15 or FxaC_2g30690), respectively.

We look forward to receiving your revised manuscript.

Kind regards,

Sara Amancio

Academic Editor

PLOS ONE

Journal Requirements:

Reviewers' comments:

Reviewer's Responses to Questions

**Comments to the Author**

1. If the authors have adequately addressed your comments raised in a previous round of review and you feel that this manuscript is now acceptable for publication, you may indicate that here to bypass the “Comments to the Author” section, enter your conflict of interest statement in the “Confidential to Editor” section, and submit your "Accept" recommendation.

Reviewer #2: All comments have been addressed

Reviewer #3: All comments have been addressed

2. Is the manuscript technically sound, and do the data support the conclusions?

Reviewer #2: Yes

Reviewer #3: Partly

3. Has the statistical analysis been performed appropriately and rigorously? 

Reviewer #2: Yes

Reviewer #3: Yes

4. Have the authors made all data underlying the findings in their manuscript fully available?

Reviewer #2: Yes

Reviewer #3: Yes

5. Is the manuscript presented in an intelligible fashion and written in standard English?

Reviewer #2: Yes

Reviewer #3: Yes

6. Review Comments to the Author

Reviewer #2: I have one last consideration: isn't the FxaC_4g15020 referred as FaMYB10-1 in ref n. 36 (Allelic Variation of MYB10 Is the Major Force Controlling Natural Variation in Skin and Flesh Color in Strawberry (Fragaria spp.) Fruit)?

Chromosomes Fvb1-1 and Fvb1-2 carry one FaMYB10 homoeolog each: FaMYB10-1 (maker-Fvb1-1-snap-gene-139.18 or FxaC_4g15020) and FaMYB10-2 (maker-Fvb1-2-snap-gene-157.15 or FxaC_2g30690), respectively.

Reviewer #3: (No Response)

7. PLOS authors have the option to publish the peer review history of their article (what does this mean?). If published, this will include your full peer review and any attached files.

Reviewer #2: **Yes: **Remo Chiozzotto

Reviewer #3: No

---

## [Author Response · Author response to Decision Letter 1]

14 May 2021

We sincerely thanks to you and the reviews for the thoughtful and helpful considerations. The name of FaMYB10 genes were revised in lines 301-304 according to the previous study [36].

---

## [Editor Report · Decision Letter 2]

20 May 2021

Expanded transcriptomic view of strawberry fruit ripening through meta-analysis

PONE-D-20-40314R2

Dear Dr. Yi,

We’re pleased to inform you that your manuscript has been judged scientifically suitable for publication and will be formally accepted for publication once it meets all outstanding technical requirements.

Kind regards,

Sara Amancio

Academic Editor

PLOS ONE
---

## [Editor Report · Acceptance letter]

21 May 2021

PONE-D-20-40314R2 

Expanded transcriptomic view of strawberry fruit ripening through meta-analysis 

Dear Dr. Yi:

I'm pleased to inform you that your manuscript has been deemed suitable for publication in PLOS ONE. Congratulations! Your manuscript is now with our production department. 

Kind regards, 

on behalf of

Prof Sara Amancio 

Academic Editor

PLOS ONE